# Genome-Wide Detection of Quantitative Trait Loci and Prediction of Candidate Genes for Seed Sugar Composition in Early Mature Soybean

**DOI:** 10.3390/ijms24043167

**Published:** 2023-02-05

**Authors:** Li Hu, Xianzhi Wang, Jiaoping Zhang, Liliana Florez-Palacios, Qijian Song, Guo-Liang Jiang

**Affiliations:** 1School of Agriculture, Yunnan University, Kunming 650091, China; 2College of Agriculture, Nanjing Agricultural University, Nanjing 210095, China; 3Department of Crop, Soil, and Environmental Sciences, University of Arkansas, Fayetteville, AR 72701, USA; 4USDA-ARS Beltsville Agricultural Research Center, Beltsville, MD 20705, USA; 5Agricultural Research Station, College of Agriculture, Virginia State University, Petersburg, VA 23806, USA

**Keywords:** *Glycine max*, soluble carbohydrates, GWAS, gene prediction

## Abstract

Seed sugar composition, mainly including fructose, glucose, sucrose, raffinose, and stachyose, is an important indicator of soybean [*Glycine max* (L.) Merr.] seed quality. However, research on soybean sugar composition is limited. To better understand the genetic architecture underlying the sugar composition in soybean seeds, we conducted a genome-wide association study (GWAS) using a population of 323 soybean germplasm accessions which were grown and evaluated under three different environments. A total of 31,245 single-nucleotide polymorphisms (SNPs) with minor allele frequencies (MAFs) ≥ 5% and missing data ≤ 10% were selected and used in the GWAS. The analysis identified 72 quantitative trait loci (QTLs) associated with individual sugars and 14 with total sugar. Ten candidate genes within the 100 Kb flanking regions of the lead SNPs across six chromosomes were significantly associated with sugar contents. According to GO and KEGG classification, eight genes were involved in the sugar metabolism in soybean and showed similar functions in Arabidopsis. The other two, located in known QTL regions associated with sugar composition, may play a role in sugar metabolism in soybean. This study advances our understanding of the genetic basis of soybean sugar composition and facilitates the identification of genes controlling this trait. The identified candidate genes will help improve seed sugar composition in soybean.

## 1. Introduction

Soybean [*G. max* (L) Merr.] is one of the major sources of human food and livestock feed in the world. Soybean seed consists of approximately 40% protein, 20% oil, and 33% carbohydrates of the dry seed weight [1,2,3,4]. Soluble sugar is an important component of carbohydrates, and it mainly comprises sucrose, stachyose, raffinose, glucose, and fructose in soybean seed [5]. As the major components of seed sugar in soybean, sucrose, stachyose, and raffinose make up 41.3–67.4%, 12.1–35.2%, and 5.2–15.8% of the total soluble sugar, respectively [6]. The proportion of sugar components in soybean seed considerably affects the quality, digestibility, and nutritional values of soy food. For soy food, such as soymilk, tofu, natto, edamame, and many other products, higher contents of sucrose, glucose, and fructose are preferred because they contribute to the favorable sweet taste and are ready to digest, while raffinose and stachyose are indigestible and cause undesirable flatulence and diarrhea [7,8].

Major quality traits of soybean seed such as protein and oil contents have been extensively investigated. Research spans from quantitative genetics to molecular mapping and candidate gene identification, especially in plant breeding and genetics [9,10,11,12,13,14]. However, to date, only a limited number of studies have been conducted to investigate the genetic architecture of sugar components using quantitative trait locus (QTL) mapping and/or genome-wide association study (GWAS). Using 149 F_2_ individuals, Maughan et al. [1] mapped ten QTLs associated with seed sucrose content on seven chromosomes (5, 7, 8, 13, 15, 19, and 20), which explained 6.1 to 12.4% of the total phenotypic variation. Kim et al. [15,16] used 115 segregating F_10_ lines and 117 F_2:10_ recombinant inbred lines (RILs) to conduct molecular mapping with simple sequence repeat (SSR) markers and detected six QTLs for oligosaccharides and sucrose on six chromosomes (2, 11, 12, 16, and 19). A major QTL on chromosome 19 accounted for 21.4% phenotypic variation, two QTLs on chromosome 12 and 16 explained 10% phenotypic variation, and remaining QTLs exhibited minor effects [15,16]. In a study with two F_2_ populations derived from the crosses of PI870139 × PI200508 and PI2435459 × PI200508, Skoneczka et al. [17] identified a major QTL on chromosome 6 responsible for low-stachyose and high-sucrose contents in PI 200508. They found that this QTL could explain 88–94% of the phenotypic variation for stachyose and 76% of the variation for sucrose content. In the study conducted by Zeng et al. [18] using 220 F_2:3_ line derived from MFS-553 × PI243545, three sucrose QTLs located on chromosomes 5, 9, and 16 were mapped and explained 46, 10, and 8% of the phenotypic variation, respectively. Wang et al. [19] used 170 F_2:3_ RILs derived from V97-3000 × V99-5089 as the mapping population and identified eleven QTLs for sugar compositions, including one for glucose, three for fructose and sucrose, and two for raffinose and stachyose. In a study with 92 F_5:7_ RILs derived from MD965722 × Spencer, Akond et al. [20] reported fourteen significant QTLs on eight chromosomes (1, 3, 6, 9, 12, 14, 15, and 16) which were associated with sugar components, including three for sucrose, seven for raffinose, and four for stachyose. A major QTL for sucrose on chromosome 15 and one QTL for stachyose on chromosome 14 explained 68 and 28% of the phenotypic variation, respectively, while altogether the seven QTLs for raffinose explained 73–76% of the phenotypic variation. In the study conducted by Salari et al. [21], four QTLs associated with sucrose and raffinose were identified in a population of 140 RILs derived from IA3023 × LD02-5585. Of the three QTLs for sucrose, two on chromosome 1 explained 22% of the phenotypic variation, and the one on chromosome 3 explained 10% of the phenotypic variation. The QTL for raffinose on chromosome 6 accounted for 7% of the phenotypic variation [21].

GWAS is regarded as a valid alternative of linkage-association analysis for its high resolution and accuracy, and it has a wide use in analysis of complex traits in soybean [22]. To date, GWAS has been utilized to study the genetic basis of important quality traits such as protein, oil, fatty acid, and amino acid contents in soybean [9,10,23,24]. For seed sugar composition in soybean, however, limited studies of GWAS have been reported, while the research has been focused mainly on phenotypic evaluation and quantitative genetic analysis of the traits [25,26]. Sui et al. [27] conducted a GWAS focusing on the sucrose content using 178 elite accessions and 33,149 single-nucleotide polymorphisms (SNPs), and ten genes within the flanking regions of the 35 significantly associated SNPs were identified. Of these genes, one had been reported previously to be associated with sucrose content, while the other nine were considered as novel genes for sucrose content. Lu et al. [28] reported a GWAS on total soluble sugar content in soybean using 278 diverse soybean accessions. They detected 84 SNPs from seventeen genes across four chromosomes to be associated with the total soluble sugar content. In addition, gene expression analysis revealed that two of these genes were positively and one was negatively associated with soluble sugar content, while six genes showed different expressions in different soybean germplasms.

To obtain a more comprehensive understanding of the genetic architecture of sugar composition in soybean, we conducted a GWAS of fructose, glucose, sucrose, raffinose, stachyose, and total sugar contents in a population of 323 germplasm accessions, which was phenotyped under three environments and genotyped with the SoySNP50K BeadChips [29]. The objectives of this study were to identify QTLs associated with sugar composition and to predict the candidate genes within the flanking regions of the peak SNP loci. The study would enhance our understanding of the genetic architecture of sugar composition and expedite the identification of genes that regulate sugar composition in soybean. It will also help in the development of markers that can be used to improve sugar composition in soybean.

## 2. Results

### 2.1. Statistics of Phenotypes

Analysis of variance showed highly significant differences among 323 germplasm accessions for all the traits investigated [26]. The contents of all individual sugars and total sugar exhibited continuous frequency distributions, which approximated to a normal distribution. Overall, the sugar contents averaged 103.6 mg g^−1^ for total sugar (70.0–140.3 mg g^−1^), 50.7 mg g^−1^ for sucrose (33.6–72.8 mg g^−1^), 40.7 mg g^−1^ for stachyose (14.7–58.3 mg g^−1^), 9.7 mg g^−1^ for raffinose (0.6–24.7 mg g^−1^), 1.7 mg g^−1^ for glucose (0.1–5.2 mg g^−1^), and 0.9 mg g^−1^ for fructose (0.0–2.9 mg g^−1^). Heritability estimates were greater than 94% for all traits, indicating that genetic effects predominantly influenced seed sugar. More detailed information of phenotypic and quantitative analysis refers to Jiang et al. [26]. 

### 2.2. Linkage Disequilibrium and Population Structure

Of 42,509 SNPs genotyped, a total of 31,245 SNPs had a minor allele frequency (MAF) ≥ 0.05 in the population and were included for a GWAS of fructose, glucose, sucrose, raffinose, stachyose, and total sugar. The mean linkage disequilibrium (LD) decay distance of the panel was ∼330 Kb when r^2^ dropped to half (Figure 1a). All the selected SNPs were used for population structure and kinship analyses. Analysis demonstrated that the first three principal components (PCs) accounted for 24.4% of the genetic variation (Figure 1b), and there was a moderate level of familial relationships within the 323 soybean germplasms (Figure 1c). 

### 2.3. Genome-Wide Association Analysis of Sugar Composition

General linear model (GLM) and mixed linear model (MLM) were employed to conduct association analysis. Principal component analysis was performed with the whole set of SNPs to capture the overall population stratification of the association panel. False discovery rate (FDR) and Type 1 error were calculated for both GLM and MLM, and the results are presented in Appendix A. Compared with the GLM that only involves population structure, the MLM that considers both population structure and kinship [30,31] showed greater control over genomic inflation, i.e., FDR and Type 1 error (Appendix A). Therefore, further analysis and subsequent results refer only to the GWAS using the MLM unless otherwise stated. The quantile-quantile plots of association analysis for all the traits are presented in Appendix A.

In total, 255 SNPs were identified to be significantly associated with the contents of individual sugars and total sugar in soybean seed. Of them, 109 SNPs for fructose were on 13 chromosomes (Figure 2a), 18 SNPs for glucose were on 4 chromosomes (Figure 2b), 50 SNPs for sucrose were on 11 chromosomes (Figure 2c), 20 SNPs for raffinose were on 8 chromosomes (Figure 2d), 25 SNPs for stachyose were on 9 chromosomes (Figure 2e), and 33 SNPs for total sugar were located on 9 soybean chromosomes (Figure 2f). The contribution of a single SNP to the phenotypic variation was 7.1–9.6% for fructose, 3.5–4.1% for glucose, 8.9–10.4% for sucrose, 10.7–11.7% for raffinose, 3.5–5.9% for stachyose, and 9.4–12.5% for total sugar. For convenience of further analysis, the significant trait-associated SNPs located in close proximity were clumped at LD r^2^ > 0.70 as described previously [32], the interval was declared as a QTL accordingly, and the SNP with the highest LOD represented the locus (lead SNP). As a result, 27, 6, 17, 11, 11, and 14 loci for fructose, glucose, sucrose, raffinose, stachyose, and total sugar were identified, respectively (Table 1). 

Among all the identified SNPs, some SNPs were associated with more than one trait. For example, the SNPs Gm02_8369052 for total sugar and Gm02_8379231 for sucrose were in the same LD block. The SNP Gm06_14414191 was associated with sucrose, raffinose, and total sugar. Two SNPs Gm09_3173391 and Gm15_49568824 were associated with both sucrose and total sugar, and two SNPs Gm12_35531777 and Gm18_7423285 were associated with stachyose and total sugar. 

### 2.4. Loci Effects and Prediction of Candidate Genes

The 100 Kb flanking each lead SNP was used as the region for candidate gene scanning. A total of 735 genes within the flanking region of 86 lead SNPs (Table 1) were found (Appendix A). Of these genes, 267 for fructose based on 27 SNPs, 16 for glucose on 6 SNPs, 135 for sucrose on 17 SNPs, 95 for raffinose on 11 SNPs, 113 for stachyose on 11 SNPs, and 109 for total sugar on 14 SNPs were predicted (Appendix A). In these regions, genes of known function in soybean or in Arabidopsis (*Arabidopsis thaliana*) related to the trait under study or genes linked or closely associated with a lead SNP located on the known QTLs were selected. Ten candidate genes on six chromosomes were identified, including four for fructose, two for sucrose, two for raffinose, one for stachyose, and one gene for total sugar (Table 1). The detailed results of candidate gene prediction for each trait are presented as follows:

#### 2.4.1. Fructose

Four candidate genes for seed fructose were identified. The known QTL Seed oligosaccharide 1-1 [15] was targeted by the SNP Gm02_39812316. It was located 4.3 Kb upstream of the transcript start site of a putative gene, *Glyma.02g212400*. There was a significant difference between lines segregating at this locus (Figure 3a). The known QTL Seed oligosaccharide 2-2 [16] was targeted by the SNP Gm06_34566459. The putative gene *Glyma.06g228000* for this locus encodes protein, sharing 84.4% similarity with the ortholog in Arabidopsis (Phytozome, http://www.phytozome.net, accessed on 27 December 2022), and it belongs to the DNAJ heat shock family. The soybean lines carrying different alleles of this locus showed a significant difference as well (Figure 3b). Two candidate genes *Glyma.16g156800* and *Glyma.16g156900* for this locus were identified close to the SNP Gm16_31661187. Both the putative genes *Glyma.16g156800* and *Glyma.16g156900* encode sucrose transport protein SUC1-related that share 80.4% and 90.3% similarity with sucrose transport protein SUC1-related in Arabidopsis (Phytozome, http://www.phytozome.net, accessed on 27 December 2022), which is involved in the sucrose metabolic process. 

#### 2.4.2. Sucrose

Two candidate genes were identified for seed sucrose. The QTL Seed oligosaccharide 1-1 [15] was led by the SNP Gm02_40701593. The putative gene *Glyma.02g218600* for this locus encodes a sucrose transporter and related protein, which shares 84.6% similarity with sucrose transporter 4 in Arabidopsis. The other putative gene for sucrose was *Glyma.03g149300*, which was led by the SNP Gm03_36427644. This gene shares 83.3% similarity with sugar transporter ERD6-like 7 in Arabidopsis and is involved in sugar–proton symport activity and carbohydrate transmembrane transporter activity pathways. 

#### 2.4.3. Raffinose

Two candidate genes were identified for seed raffinose. The putative gene *Glyma.05g003900* led by the SNP Gm05_317349 encodes raffinose synthase family protein, which shares 81.7% similarity with the ortholog in Arabidopsis. The other putative gene *Glyma.15g142400* encodes glycosyl hydrolase superfamily protein sharing 78.5% similarity with the ortholog in Arabidopsis, which is involved in glucan endo-1,3-beta-D-glucosidase activity.

#### 2.4.4. Stachyose 

One gene for stachyose was detected. The putative gene *Glyma.03g002100* led by the SNP Gm03_212961 encodes glycosyl hydrolase superfamily protein and major facilitator superfamily protein, which shares 83.6% similarity with the ortholog in Arabidopsis and is involved in the carbohydrate metabolic process. 

#### 2.4.5. Total Sugar

For total sugar, one gene *Glyma.05g168600* led by the SNP Gm05_35918853 was identified. This putative gene encodes protein, sharing 81% similarity with the ortholog UDP-glucose:glycoprotein glucosyltransferase (HUGT) in Arabidopsis.

## 3. Discussion

Relative kinship and population structure are considered as two major confounding factors that may lead to spurious results in an association study [30]. Compared with the GLM, the MLM takes both familial relatedness and population structure into account, which has been proven as an effective way to control genomic inflation and has been widely used in GWAS for various complex traits in multiple plant species [30,35,36,37]. Our previous study has also shown that a marker-based principal analysis is sufficiently flexible to generate a trait-specific population structure that optimizes the model fitness [10,32,38]. In this study, the MLM showed a better control of genomic inflation than the GLM with population structure (Supplementary File Appendix A), indicating that the MLM was more appropriate than the GLM for the GWAS of seed sugar composition. 

As reported, a total of thirty-seven QTLs for sucrose concentration and fifteen QTLs for oligosaccharide concentration have been identified so far (http://www.soybase.org, accessed on 27 December 2022). In addition to linkage analysis, GWAS has been proven to be powerful in the identification of loci that associated with numerous traits in crop plants such as soybean [39,40,41,42]. In this study, a total of 86 SNPs were detected to be associated with sugar composition, including 27, 6, 17, 11, 11, and 14 SNPs for fructose, glucose, sucrose, raffinose, stachyose, and total sugar content, respectively. Some of the SNPs identified were overlapped with the ones previously reported. For example, the SNPs Gm02_39812316 and Gm02_37116580 for fructose were identified to be located within the marker interval of a previously reported QTL, Seed oligosaccharide 1-1 [15], which was located in the physical position of 31.5–42.0 Mb on chromosome 2 [33]. The SNP Gm05_35918853 for total sugar was located within 33.5–37.5 Mb on chromosome 5 in which a QTL associated with soluble sugar contents exists as well [33]. Three SNPs Gm06_34566459, Gm06_36593777, and Gm06_38660086 for fructose were located at 28.0–40.5 Mb on chromosome 6, wherein a QTL is associated with soluble sugar contents [33]. In addition, two QTLs Seed oligosaccharide 2-2 [16] and qSuc_06 (BREC12) [34] are also located in this region. The SNP Gm03_212961 for stachyose was 20 Kb away from the marker Gm03_192792, which has been identified to be associated with sucrose [21]. Because raffinose and stachyose are derived from sucrose, possibly these two SNPs point to the same QTL. The SNP Gm15_49568824 associated with both sucrose and total sugar content was 293 Kb away from the SNP rs49892059 (Gm15_49892059), which was the peak marker of a QTL for soluble sugar content in a previous study [28]. 

For the present, hundreds of candidate genes have been known to be involved in the metabolism of sugars in soybean (http://www.soybase.org, accessed on 27 December 2022, and http://www.phytozome.net, accessed on 27 December 2022). GWAS is the primary choice of methods that can be used for excavating and evaluating major genes due to its high resolution [22]. In this study, a total of 735 genes were found in the 100 Kb flanking regions of the eighty-six SNPs, and ten candidate genes were predicted to be possibly involved in the metabolisms of sugars in soybean. According to GO and KEGG classification, eight genes were involved in the sugar metabolism in soybean and showed similar function in Arabidopsis, including *Glyma.02g218600*, *Glyma.03g002100*, *Glyma.03g149300*, *Glyma.05g003900*, *Glyma.05g168600*, *Glyma.15g142400*, *Glyma.16g156800*, and *Glyma.16g156900*. The putative genes *Glyma.02g218600* and *Glyma.03g149300* for sucrose encode sucrose transporter and related proteins and sugar transporter ERD6-like 7, respectively, which is directly involved in sucrose transport and sugar transport. The putative gene *Glyma.05g003900* for raffinose encodes raffinose synthase family protein. The putative gene *Glyma.05g168600* for total sugar encodes UDP-glucose:glycoprotein glucosyltransferase. The putative genes *Glyma.03g002100* and *Glyma.15g142400* for stachyose and raffinose encode glycosyl hydrolase superfamily protein, and this protein has been proven to be involved in the metabolism of cell wall polysaccharides in plants [43,44]. The putative genes *Glyma.16g156800* and *Glyma.16g156900* for fructose encode sucrose transport protein SUC1-related, which is directly involved in sucrose transport. 

As well known, sugar metabolism is the most basic metabolism in all creatures, and the components or constituents of sugar are interchangeable. Fructose and glucose compose sucrose, while raffinose and stachyose are derived from sucrose, and so on. Therefore, it is assumable that the identified genes could be associated with different sugar compositions. Other two genes *Glyma.02g212400* and *Glyma.06g228000* for fructose were targeted to the known QTLs Seed oligosaccharide 1-1 and Seed oligosaccharide 2-2. The putative gene *Glyma.02g212400* encodes cyclin, sharing 72% similarity with CYCLIN-D2-1 in Arabidopsis (http://www.phytozome.net, accessed on 27 December 2022) and GO:0009744, which shows that this gene responds to sucrose stimulus in the biological process of Arabidopsis. Previous research also showed that sucrose seemed to differentially regulate D-type cyclin expression in Arabidopsis regarding both kinetics and the rate of induction [45]. It suggests that this gene might play an important role in the sugar composition of soybean seeds. The putative gene *Glyma.06g228000* encodes DNAJ heat shock family protein, which has been proven to be involved in a variety of essential cellular processes, including de novo protein folding, translocation of polypeptides across cellular membranes, and degradation of misfolded proteins [46]. Therefore, it is worth a try to explore whether *Glyma.06g228000* plays a role in the sugar metabolism of soybean seeds. The identified candidate genes associated with sugar contents are bases for further gene function verification and the improvement of seed sugar composition in soybean. 

## 4. Materials and Methods

### 4.1. Plant Materials

A total of 323 plant introductions (PIs) of early maturity groups 0 and 00, obtained from the United States Department of Agriculture (USDA) Soybean Germplasm Collection, were used in this study. All the PIs were planted in single-row plots in a randomized complete block design with three replications at three environments: Aurora (44°18′ N and 96°40′ W, 2011), Brookings (44°27′ N and 96°47′ W, 2012), and Watertown (45°06′ N and 97°05′ W, 2012) in South Dakota, USA. The detailed information of the 323 PIs, field experiments, and seed sampling were given in our previous publication [26].

### 4.2. Phenotypic Evaluation and Statistical Analysis

As described previously [26], sugar sample preparation, extraction, and analysis were conducted in the Soybean Breeding and Genetics Laboratory at the University of Arkansas using the HPLC method [25]. Briefly, glucose, fructose, sucrose, raffinose, and stachyose in the extracts were determined and quantified based on the retention time and the regression curve established for each of five standard sugars. Analysis of variance, frequency distribution, heritability estimation, and genotypic and phenotypic correlation were published in our previous report [26]. 

### 4.3. Genotyping and Quality Control

The Illumina Infinium SoySNP50K BeadChips were used to genotype the population as described by Song et al. [29,47], and 42,509 SNPs were identified to show a call success rate of 85% or greater. The SNPs with a MAF < 5% in the population were excluded from further analysis for quality control. Finally, a total of 31,245 SNPs were used for the GWAS. 

### 4.4. Linkage Disequilibrium (LD) Estimation

Pairwise LD between markers was calculated as the squared correlation coefficient (r^2^) of alleles using PopLDdecay [48]. The LD decay rate of the population was measured as the chromosomal distance where the average r^2^ dropped to half of its maximum value [35].

### 4.5. Genome-Wide Association Analysis

To minimize the effects of environmental variation, best linear unbiased predictions (BLUPs) of genetic effect for each line were calculated using the R package lme4 [49] in the same model as described for phenotypic trait. The BLUPs were then used for association analysis. Two models, GLM with principal component analysis and MLM involving both principal component analysis and kinship, were implemented in the R package called Genomic Association and Prediction Integrated Tool (GAPIT) [31,50]. The significance threshold for SNP-trait associations was determined by false discovery rate (q value) *p* < 0.001.

### 4.6. Prediction of Candidate Genes 

Genes within the 100 Kb flanking region of the lead SNPs were selected as the source of candidate genes, and the functions of these genes were annotated according to the soybean reference genome (Wm82.a2.v1, http://www.soybase.org, accessed on 27 December 2022) and Arabidopsis genome assembly (Arabidopsis thaliana Araport11, http://www.phytozome.net, accessed on 27 December 2022). Candidate genes were predicted with reference to the following preferences: (i) genes of known function in soybean related to the trait under study, (ii) genes with function-known orthologs in Arabidopsis related to the trait under study, and (iii) genes linked or closely associated with a lead SNP located within the known QTL regions.

## 5. Conclusions

To better understand the genetic architecture underlying the sugar composition in soybean, a GWAS was conducted for the contents of seed fructose, glucose, sucrose, raffinose, stachyose, and total sugar in a population of 323 soybean germplasm accessions. All the germplasm accessions were evaluated under three environments and were genotyped with 42,509 SNP markers, of which 31,245 had a minor allele frequency (MAF) ≥ 5% and missing data ≤ 10% and were finally used for the GWAS. In total, 86 QTLs were identified to be associated with individual sugars and total sugar. Within the 100 Kb flanking regions of the lead SNPs across six chromosomes, ten candidate genes were identified to be significantly associated with sugar composition. Eight of these genes were involved in sugar metabolism in soybean and showed similar functions in Arabidopsis, and the other two located in the known QTL regions may play a role in sugar metabolism in soybean. These identified candidate genes will lay a foundation for further gene function verification and the improvement of seed sugar composition in soybean. This study will not only advance our understanding of the genetic basis of soybean sugar composition but will also facilitate the identification of genes controlling this trait.

## Figures and Tables

**Figure 1 ijms-24-03167-f001:**
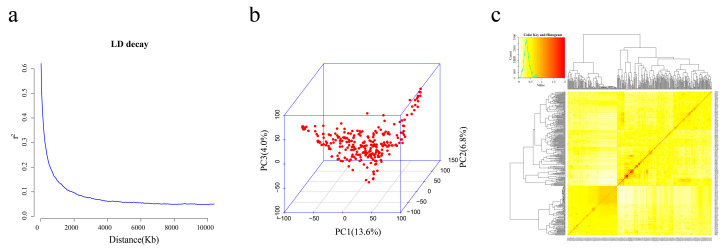
Linkage disequilibrium and population structure of the GWAS panel. (**a**) Linkage disequilibrium (LD) decay of the association analysis panel. (**b**) The first three PCs of 31,245 SNPs used in the GWAS. (**c**) A heatmap of the kinship matrix of the 323 soybean accessions.

**Figure 2 ijms-24-03167-f002:**
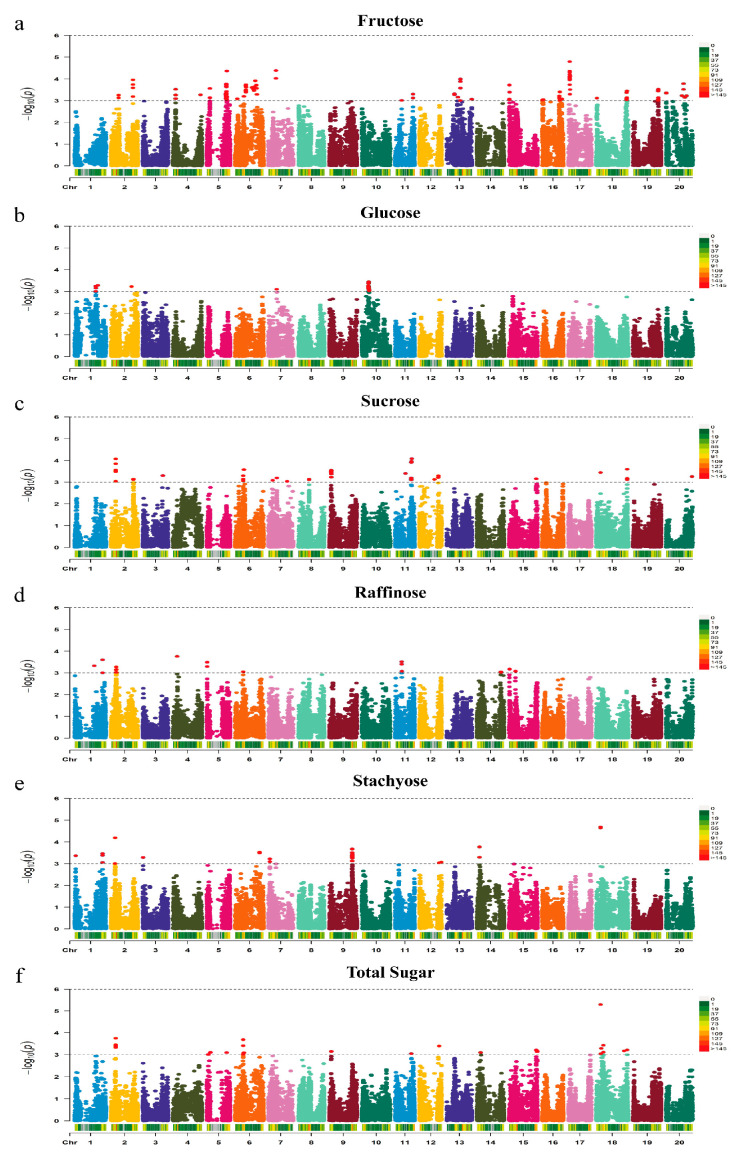
Manhattan plot of GWAS for Fructose (**a**), Glucose (**b**), Sucrose (**c**), Raffinose (**d**), Stachyose (**e**), and Total Sugar content (**f**) from 323 soybean accessions. The significant trait-associated SNPs (−log_10_(P) >3) are distinguished by the threshold line and colored in red (color Figure online).

**Figure 3 ijms-24-03167-f003:**
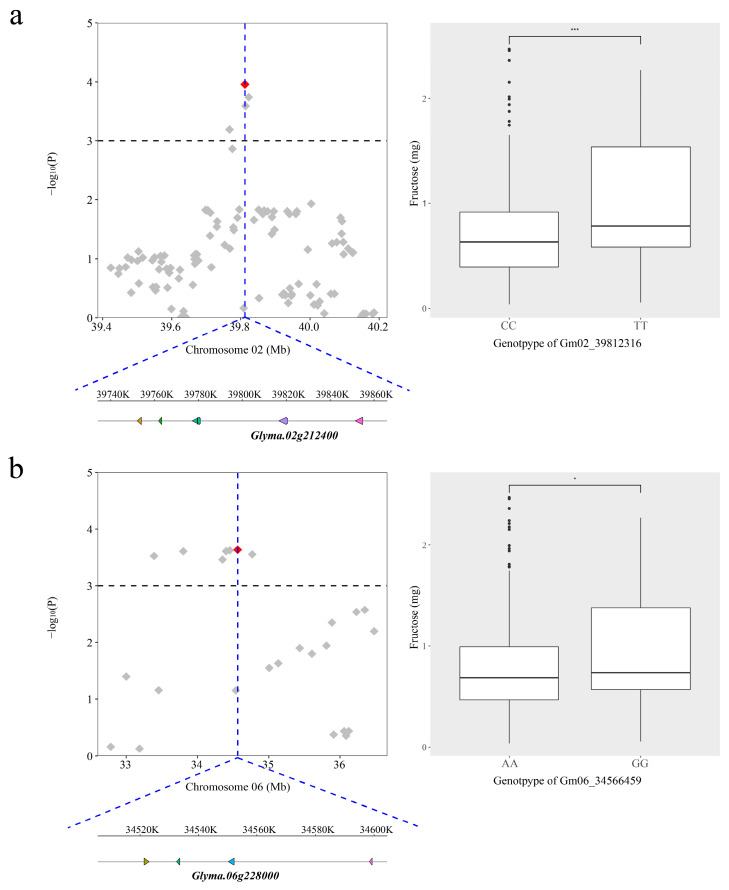
Candidate genes for lead SNP associated with fructose on Gm02 and Gm06, and phenotypic difference between the lines carrying different alleles. (**a**). The SNP Gm02_39812316. (**b**). The SNP Gm06_34566459. The top sections of the left panels show regions on each side of the lead SNP, whose position is indicated by a vertical blue dashed line. Negative log10 transformed P values of SNPs from the MLM are plotted on the vertical axis. Significant threshold is indicated as the gray dashed line at −log_10_(P) = 3. The bottom sections of the left panels show putative genes within 100 Kb adjacent regions on each side of the lead SNP. Candidate genes are indicated by their names. The boxplots of the right panels show the distribution of fructose over four environments for each locus allele. Significance of ANOVA: * *p* < 0.05 and *** *p* < 0.001.

**Table 1 ijms-24-03167-t001:** Peak SNPs significantly associated with fructose, glucose, sucrose, raffinose, stachyose, and total sugar, as well as predicted genes for each trait at 100 Kb genome regions. Genes annotated in Glyma.a2.v1 in SoyBase (https://www.soybase.org, accessed on 27 December 2022) and Arabidopsis thaliana Araport11 in Phytozome (http://www.phytozome.net, accessed on 27 December 2022) with related functions were used as the source of candidate genes. QTN: quantitative trait nucleotide.

Trait	SNP ID	Alleles	MAF	−log_10_(P)	Known QTLs and QTNs	Candidate Gene	Annotation
Fructose	Gm02_13523639	G:T	0.36	3.25			
	Gm02_39812316	T:C	0.45	3.96	Seed oligosaccharide 1-1 [15]	Glyma.02g212400	CYCLIN-D4-1-RELATED
					Gm02:31.5–42.0 Mb [33]		
	Gm04_4733978	G:A	0.42	3.52			
	Gm04_49873994	C:T	0.48	3.27			
	Gm05_5397251	T:C	0.14	3.56			
	Gm05_36311870	A:G	0.37	4.36			
	Gm06_19321023	A:C	0.45	3.69			
	Gm06_34566459	G:A	0.49	3.63	Seed oligosaccharide 2-2 [16]	Glyma.06g228000	DNAJ heat shock family protein
					Gm06:28.0–40.5 Mb [33]		
	Gm06_36593777	T:G	0.50	3.91	qSuc_06 (BREC12) [34]		
					Gm06:28.0–40.5 Mb [33]		
	Gm06_38660086	G:A	0.50	3.71	Gm06:28.0–40.5 Mb [33]		
	Gm07_13530340	G:A	0.50	4.39			
	Gm11_11249766	G:A	0.26	3.01			
	Gm11_32043302	C:A	0.26	3.30			
	Gm13_13865523	A:G	0.08	3.31			
	Gm13_24748152	C:T	0.50	3.99			
	Gm13_45487373	C:T	0.21	3.07			
	Gm15_548936	T:G	0.28	3.72			
	Gm16_97490	G:A	0.39	3.03			
	Gm16_1358677	G:A	0.32	3.04	Seed sucrose 3–5, Seed oligosaccharide 2–5 [15]		
					Gm16:1.5–2.5 Mb [33]		
	Gm16_31661187	C:T	0.29	3.40		Glyma.16g156800	Sucrose transport protein SUC1-related
						Glyma.16g156900	Sucrose transport protein SUC1-related
	Gm17_2425050	G:T	0.37	4.30			
	Gm17_3094261	C:T	0.38	4.13			
	Gm18_949928	C:T	0.20	3.12			
	Gm18_55715030	A:G	0.33	3.44			
	Gm19_45790916	G:A	0.19	3.52			
	Gm20_58755	C:T	0.41	3.36			
	Gm20_31554795	C:T	0.28	3.78			
Glucose	Gm01_38064415	T:G	0.13	3.24			
	Gm01_42155458	C:A	0.11	3.28			
	Gm02_37116580	T:C	0.17	3.22	Gm02:31.5–42.0 Mb [33]		
	Gm07_14920947	A:G	0.30	3.10			
	Gm10_11710604	G:A	0.25	3.43			
	Gm10_12143793	T:C	0.28	3.44			
Sucrose	Gm02_8379231	G:A	0.09	4.07			
	Gm02_40701593	A:G	0.36	3.14	Seed oligosaccharide 1–1 [15]	Glyma.02g218600	Sucrose transporter and related proteins
					Gm02:31.5–42.0 Mb [33]		
	Gm03_36427644	C:T	0.31	3.30		Glyma.03g149300	Sugar transporter ERD6-like 7
	Gm06_14414191	G:A	0.15	3.29			
	Gm06_15825296	T:C	0.26	3.58			
	Gm07_7692973	C:T	0.08	3.08			
	Gm07_15235896	A:C	0.23	3.19			
	Gm07_34157286	C:T	0.40	3.04			
	Gm08_20126261	C:T	0.07	3.14			
	Gm09_3173391	A:C	0.13	3.54	Seed sucrose 4–2 [18]		
					Gm09:3.0–6.0 Mb [33]		
	Gm11_18321103	C:T	0.49	3.40			
	Gm11_29223742	C:T	0.06	3.95			
	Gm12_34184311	A:G	0.07	3.30			
	Gm15_49568824	C:A	0.09	3.16			
	Gm18_7655919	C:T	0.05	3.44			
	Gm18_56084992	C:A	0.15	3.59			
	Gm20_47044176	C:A	0.15	3.26			
Raffinose	Gm01_35419697	G:A	0.11	3.33			
	Gm01_50576284	T:G	0.12	3.60			
	Gm02_9220693	A:C	0.25	3.27			
	Gm02_9675440	C:T	0.43	3.16			
	Gm04_7246268	T:C	0.12	3.76			
	Gm05_317349	C:T	0.15	3.49		Glyma.05g003900	Raffinose synthase family protein
	Gm06_14414191	G:A	0.15	3.05			
	Gm11_11311013	C:T	0.23	3.52			
	Gm14_43573428	G:A	0.50	3.05			
	Gm15_741109	A:C	0.49	3.17			
	Gm15_11667788	A:G	0.10	3.08		Glyma.15g142400	Glycosyl hydrolase superfamily protein
Stachyose	Gm01_1675532	T:C	0.09	3.36			
	Gm01_50504115	C:T	0.28	3.46			
	Gm02_7204486	T:C	0.22	4.19			
	Gm03_212961	C:T	0.44	3.28	Gm03_192792 [21]	Glyma.03g002100	Glycosyl hydrolase superfamily protein
	Gm06_44669377	T:C	0.47	3.53			
	Gm07_2535137	T:G	0.15	3.22			
	Gm09_41499208	A:C	0.15	3.68	Seed arabinose plus galactose 1–2galactose [34]		
					Gm09:38.0–42.0 Mb [33]		
	Gm12_35531777	G:A	0.13	3.04			
	Gm12_39385435	T:C	0.25	3.07			
	Gm14_4487163	T:C	0.08	3.77			
	Gm18_7423285	A:G	0.06	4.69			
Total sugar	Gm02_8369052	G:A	0.09	3.75			
	Gm05_2353434	C:T	0.06	3.00			
	Gm05_6149022	G:A	0.29	3.10			
	Gm05_35918853	T:C	0.07	3.09	Gm05:33.5–37.5 Mb [33]	Glyma.05g168600	UDP-glucose:glycoprotein glucosyltransferase
	Gm06_14414191	G:A	0.15	3.68			
	Gm09_3173391	A:C	0.13	3.14	Seed sucrose 4–2 [18]		
					Gm09:3.0–6.0 Mb [33]		
	Gm11_28687660	T:C	0.05	3.05			
	Gm12_35531777	G:A	0.13	3.39			
	Gm14_6503731	C:T	0.08	3.10			
	Gm15_49568824	C:A	0.09	3.21	SNP rs49892059 [28]		
	Gm15_51132707	G:T	0.09	3.16			
	Gm18_7423285	A:G	0.06	5.30			
	Gm18_12706792	G:T	0.19	3.43			
	Gm18_56065506	C:T	0.09	3.21			

## Data Availability

The genotyping data of all 323 germplasm accesses with SoySNP50K Bead-Chips [29,47] is available at http://www.soybase.org, accessed on 27 December 2022.

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
