# Peer review of "Genome-Wide Detection of Quantitative Trait Loci and Prediction of Candidate Genes for Seed Sugar Composition in Early Mature Soybean"

_ijms, 2023, doi:10.3390/ijms24043167_

Round 1

Reviewer 1 Report

The work by  Hu et al., advances our understating the genetic architecture underlying the sugar composition in soybean seeds, by conducted a genome-wide association study (GWAS) for a population of 323 soybean germplasm accessions which were grown and evaluated under three different environments. The report is well written and condensed, as well as technically appropriate for International Journal of Molecular Sciences. However, before being able to recommend acceptance, I invite authors to address the following amendments:

-In the introduction part: In line no. 31: " Glycine max " it should be "G. max", please correct it.

-In the introduction part: In line no. 66: please add the full name of this abbreviation "RILs " it should be  "Recombinant Inbred Lines (RILs )", please correct it.

-In the results part:  Please improve the quality and clarity of the figure 1.

-In the results part:  Line no. 133: please add the full name for these two abbreviations "GLM and MLM" it should be "general linear model (GLM) and mixed linear model (MLM)", the full name of the abbreviation should be written at the first mention of it in the manuscript.

-The arrangement of photos and tables in the manuscript must be consideration, the priority of placing figures or tables must be based on where they are mentioned in the manuscript, so you should place Figure 2 before Table 1.

-Please add the name of this abbreviation "QTN" in table 1 in the table legend, it should be QTN: quantitative trait nucleotide.

-In table 1: "(PAN et al., 2022)" the author name in reference it should be "Pan et al., 2022", please correct it.

- In line no. 256, 258 and 260:  "(PAN et al., 2022)" the author name in reference it should be "Pan et al., 2022", please correct it.

-In line no. 24, 177, 190, 196, 212, 214, 220, 222, 227, 232, 276, 296, 297, 299 and 352, : "Arabidopsis" it should be full name and italic "Arabidopsis thaliana", or you can add the plant name without italic letters "Arabidopsis".

-In line no. 190 and 197: Please adjust the font size "http://www.phytozome.net", please correct it.

-In line no. 246: Please adjust the font size "(http://www.soybase.org) ", please correct it.

-In line no. 270 and 271: Please adjust the font size "(http://www.soybase.org) and http://www.phytozome.net", please correct it.

-In line no. 296: Please adjust the font size "http://www.phytozome.net and GO:0009744", please correct it.

-In line no. 349: Please adjust the font size "(http://www.soybase.org) and http://www.phytozome.net", please correct it.

-In line no. 441 and 442: please follow the instruction of authors when you wright the reference, for-example the year of publication "2015" it should be "(2015)", please correct it.

Author Response

Response: Thanks so much for your positive comments and constructive suggestions. We have made all changes as suggested.

-In the introduction part: In line no. 31: " Glycine max " it should be "G. max", please correct it.

Yes, changed.

-In the introduction part: In line no. 66: please add the full name of this abbreviation "RILs " it should be  "Recombinant Inbred Lines (RILs )", please correct it.

Changed.

-In the results part:  Please improve the quality and clarity of the figure 1.

Changed.

-In the results part:  Line no. 133: please add the full name for these two abbreviations "GLM and MLM" it should be "general linear model (GLM) and mixed linear model (MLM)", the full name of the abbreviation should be written at the first mention of it in the manuscript.

Changed.

-The arrangement of photos and tables in the manuscript must be consideration, the priority of placing figures or tables must be based on where they are mentioned in the manuscript, so you should place Figure 2 before Table 1.

Changed.

-Please add the name of this abbreviation "QTN" in table 1 in the table legend, it should be QTN: quantitative trait nucleotide.

Changed.

-In table 1: "(PAN et al., 2022)" the author name in reference it should be "Pan et al., 2022", please correct it.

Changed.

- In line no. 256, 258 and 260:  "(PAN et al., 2022)" the author name in reference it should be "Pan et al., 2022", please correct it.

Changed for all.

-In line no. 24, 177, 190, 196, 212, 214, 220, 222, 227, 232, 276, 296, 297, 299 and 352, : "Arabidopsis" it should be full name and italic "Arabidopsis thaliana", or you can add the plant name without italic letters "Arabidopsis".

Changed as suggested.

-In line no. 190 and 197: Please adjust the font size "http://www.phytozome.net", please correct it.

Changed.

-In line no. 246: Please adjust the font size "(http://www.soybase.org) ", please correct it.

Changed.

-In line no. 270 and 271: Please adjust the font size "(http://www.soybase.org) and http://www.phytozome.net", please correct it.

Changed.

-In line no. 296: Please adjust the font size "http://www.phytozome.net and GO:0009744", please correct it.

Changed.

-In line no. 349: Please adjust the font size "(http://www.soybase.org) and http://www.phytozome.net", please correct it.

Changed.

-In line no. 441 and 442: please follow the instruction of authors when you wright the reference, for-example the year of publication "2015" it should be "(2015)", please correct it.

Changed. Once again, thank you.

Reviewer 2 Report

The work brings a novel theme and very well described and discussed by the authors. however, minor corrections are suggested throughout the text. Attached file with suggestions to be corrected in the text.

Author Response

Thanks so much for your positive comments and constructive suggestions. We have made all changes as suggested (please see the manuscript for detail). For the point on moving the Supplementary file Figure S2 after the paragraph to Figure 2, we think the original place should be appropriate since Figure S2 just presents the quantile-quantile plots results of association analysis for all the traits, while Figure 2 presents the Manhattan plot of GWAS for traits. According to your suggestion, a conclusion paragraph has been added. Other typos or minor errors have been also corrected. Once again, Thank you.